# Characteristics of Extracellular Vesicles from a High-Grade Serous Ovarian Cancer Cell Line Derived from a Platinum-Resistant Patient as a Potential Tool for Aiding the Prediction of Responses to Chemotherapy

**DOI:** 10.3390/ph16060907

**Published:** 2023-06-20

**Authors:** Katarina Černe, Nuša Kelhar, Nataša Resnik, Maruša Herzog, Lana Vodnik, Peter Veranič, Borut Kobal

**Affiliations:** 1Institute of Pharmacology and Experimental Toxicology, Faculty of Medicine, University of Ljubljana, SI-1000 Ljubljana, Slovenia; nusa.kelhar@gmail.com (N.K.); vodniklana@gmail.com (L.V.); 2Institute of Cell Biology, Faculty of Medicine, University of Ljubljana, SI-1000 Ljubljana, Slovenia; natasa.resnik@mf.uni-lj.si (N.R.); peter.veranic@mf.uni-lj.si (P.V.); 3Division of Gynecology and Obstetrics, University Medical Centre Ljubljana, SI-1000 Ljubljana, Slovenia; marusa.herzog@kclj.si (M.H.); borut.kobal@kclj.si (B.K.); 4Department of Gynecology and Obstetrics, Faculty of Medicine, University Ljubljana, SI-1000 Ljubljana, Slovenia

**Keywords:** cell lines, chemoresistance, EpCAM, extracellular vesicles, HGSOC

## Abstract

Platinum-resistant high-grade serous ovarian cancer (HGSOC) is invariably a fatal disease. A central goal of ovarian cancer research is therefore to develop new strategies to overcome platinum resistance. Treatment is thus moving towards personalized therapy. However, validated molecular biomarkers that predict patients’ risk of developing platinum resistance are still lacking. Extracellular vesicles (EVs) are promising candidate biomarkers. EpCAM-specific EVs are largely unexplored biomarkers for predicting chemoresistance. Using transmission electron microscopy, nanoparticle tracking analysis and flow cytometry, we compared the characteristics of EVs released from a cell line derived from a clinically confirmed cisplatin-resistant patient (OAW28) and EVs released from two cell lines from tumors sensitive to platinum-based chemotherapy (PEO1 and OAW42). We demonstrated that EVs released from the HGSOC cell line of chemoresistant patients exhibited greater size heterogeneity, a larger proportion of medium/large (>200 nm) Evs and a higher number of released EpCAM-positive EVs of different sizes, although the expression of EpCAM was predominant in EVs larger than 400 nm. We also found a strong positive correlation between the concentration of EpCAM-positive EVs and the expression of cellular EpCAM. These results may contribute to the prediction of platinum resistance in the future, although they should first be validated in clinical samples.

## 1. Introduction

Ovarian cancer (OC), despite its infrequent incidence, is the most lethal malignancy of the female reproductive system, with most patients presenting with advanced stage tumors (FIGO stage III and IV) [1]. Although there has been an overall decreasing trend of incidence and mortality of OC over the past decade, a substantial increase in incidence has been observed in younger females [2]. In addition to the late detection of the disease, the main reason for the poor prognosis is resistance to pharmacotherapy. Despite novel targeted drugs such as poly (ADP-ribose) polymerase inhibitors (PARPis), carboplatin with added paclitaxel remains the standard first-line treatment for advanced OC. Moreover, PARPis are indicated for maintenance treatment in the first line and relapsed settings of patients who have responded to platinum-based medicine [3,4]. Additionally, despite the introduction of targeted therapy, there has been only a marginal improvement in the overall survival rate of OC patients since the introduction of cisplatin [5]. Since platinum-resistant OC is invariably a fatal disease, a central goal in OC research is the development of novel strategies to overcome platinum resistance. The treatment of OC is therefore moving toward personalized therapy. However, the selection criteria are not entirely clear since there are currently no validated molecular predictive biomarkers for platinum resistance [4].

One of the major challenges in the search for clinical biomarkers for OC is that OC is not a single disease but rather a myriad of different malignancies with different targets that share a common anatomical location at presentation. In our article, therefore, we have focused on the high-grade serous subtype of ovarian cancer (HGSOC), being the dominant and most lethal subtype of OC [6].

In recent years, tumor-derived extracellular vesicles (EVs) have attracted considerable attention as promising sources of minimally invasive biomarkers, including for OC [7,8,9,10]. According to the International Society of Extracellular Vesicles (ISEV), EV is an appropriate term for heterogeneous populations of cell-derived nanoscale vesicles released under physiological and pathological conditions, which can be isolated from cell culture supernatant or body fluids [11]. Given their important role in cell-to-cell communication [12], it is not surprising that EVs’ cargo, concentration and size have been shown to be unique in tumor states. EVs are therefore ideal candidates for aiding the prediction of chemoresistance to platinum-based medicines. Although, among published studies, those showing that the total concentration of OC-derived EVs has the potential to provide information on diagnosis predominate [13,14,15], the results of some studies do not confirm this [8,16], raising the question of whether the determination of EV concentration alone is sufficient for OC diagnosis. Apart from one preliminary study that concluded that the determination of the protein content of EVs could be useful in assessing patients’ response to therapy, we did not find any published study on the importance of determining the total EV concentration as a predictive biomarker in OC [17].

Among specific EV cargos, particular attention has been paid to the epithelial cellular adhesion molecule (EpCAM) [7,8,15,18,19]. Its expression is upregulated in solid epithelial cancer and stem cells. EpCAM is overexpressed in ~68% of patients with serous OC [20,21]. EpCAM can be found in disseminated tumor cells, circulating tumor cells (CTC) and EVs [22,23,24]. To our surprise, the relevance of the analysis of EVs in predicting chemoresistance has been largely unexplored. Research on EpCAM-specific EVs might also shed light on the contradictory results of the immunohistochemical evaluation of EpCAM overexpression in ovarian tumors in relation to its prognostic value, in terms of overall survival and response to chemotherapy [20,21,25,26]. Moreover, the determination of EpCAM-specific EVs might be helpful in evaluating the potential of therapeutic strategies targeting EpCAM and could explain the disparate results sometimes obtained concerning the effectiveness of EpCAM-based medicine [27,28]. Another promising method for predicting chemoresistance is analysis of circulating CTCs, which represent tumor cells that migrate into the blood stream. Because of the importance of EpCAM for the enrichment of CTCs, which is highly challenging due to their rarity, EpCAM-specific EVs can be useful in the investigation of changes in the EpCAM profile over time, which has been observed in patient samples [22,29,30].

Patients’ body fluids, especially blood, as a source of EVs for the study of biomarkers, have several limitations, most notably the extremely high heterogeneity of vesicles and the low proportion of EVs of tumor origin [31,32]. EVs isolated from the conditioned media of relevant cultured cells can therefore elucidate these shortcomings. Moreover, the use of well-characterized cancer cell lines to model HGSOC in vitro can be particularly valuable in the era of personalized therapy. They at least increase the likelihood that the conclusions reached in an in vitro setting will be transferable to a clinical one. Unfortunately, many of the top cell lines in terms of publication figures poorly recapitulate the genetic features of HGSOC [33,34,35]. In contrast, the most suitable cell lines for an HGSOC model (e.g., KURAMOCHI; COV362 and OAW28) are rarely used in laboratories [33]. An additional problem in cell-line models is a lack of patient-derived cell lines of chemoresistant HGSOC. The majority of cell lines used to represent platinum-resistant diseases have been generated in vitro by exposing the cells to platinum-based medicines. The type of drug exposure used to elicit the resistant phenotype is therefore unrealistic, and, as a consequence, the mechanism is likely to be different from the mechanisms acting within a patient’s body [6].

In order to identify the characteristics of EVs that have the potential to predict resistance to platinum-based medicine in HGSOC, we compared EVs released from a cell line derived from a clinically confirmed chemoresistant patient and EVs released from two cell lines from tumors sensitive to platinum-based chemotherapy. We demonstrated that the EV concentration did not differ significantly, although a significantly higher concentration of EpCAM-specific EVs and a higher proportion of medium/large particles were typical of the chemoresistant HGSOC cell line. A strong correlation between EpCAM-specific EV concentration and cell membrane EpCAM expression supports the use of EpCAM-specific EVs as cellular surrogates for EpCAM as a biomarker. Although the results of the current study should be validated in clinical samples, they reveal novel characteristics of EVs that may have the potential to aid the prediction of platinum resistant HGSOC and thereby enable the selection of personalized therapy.

## 2. Results

EVs were isolated from the conditioned media of three serous OC cell lines using a commercial reagent. One cell line was derived from a clinically confirmed cisplatin-chemoresistant patient (OAW28), and the other two were derived from patients with tumors sensitive to platinum-based chemotherapy (PEO1, OAW42). A graphic presentation of the methods is shown in Figure 1. We first confirmed the presence of EVs in isolated samples by transmission electron microscopy (TEM). The micrographs showed that EVs were successfully isolated from the conditioned media of all three cell lines (Figure 2A–C). We then determined the EVs’ characteristics by different complementary methods to identify possible differences between EVs released from cells derived from chemoresistant tumors and those released from cells derived from sensitive tumors. We used the following methods: (1) TEM, (2) nanoparticle tracking analysis in scatter mode (S-NTA) and (3) fluorescence-triggered flow cytometry (FT-FCM). The results of the EVs’ concentration were normalized by the number of viable cells. The enumeration of EVs by FT-FCM was performed using volumetric measurement (events/mL). EpCAM-positive vesicles were also specifically isolated using anti-EpCAM magnetic beads and analyzed by bead-based flow cytometry (BB-FCM). Additionally, the expression of EpCAM on cell membranes was analyzed using FCM.

### 2.1. Size Distribution of EVs: OC Cells Derived from the Clinically Confirmed Chemoresistant Patient Released More Medium/Large EVs Than OC Cells from Patients with Tumors Sensitive to Platinum-Based Chemotherapy

TEM analysis showed that isolates from conditioned media of OAV28, PEO1 and OAW42 cells consisted of EVs with a spherical shape (Figure 2A–C). Variations in the size of EVs between cell types were observed.

In order to define further the size distribution of EVs, we used S-NTA. S-NTA is the appropriate method for determining size distribution because it is based on the tracking of individual particles. Since S-NTA is not specific for EVs and detects any structure that scatters light, we refer to ‘‘particles’’ rather than ‘‘vesicles’’ in this context. The size distribution of particles from all three cell lines appeared non-symmetrical, with the highest peak ranging between approximately 100 to 140 nm in diameter and several peaks with multiple diameters greater than 200 nm (Figure 3A–C). The modal diameter of OAW28-derived EVs (134.8 ± 6.3 nm) was significantly larger than that of PEO1-derived EVs (107.3 ± 3.4 nm; *p* = 0.007), although no significant difference in modal diameter was found in comparison to OAW42-derived EVs (137.3 ± 4.2 nm; Figure 3D). On the other hand, NTA revealed a significantly larger mean diameter of EVs from OAW28 cells (234.4 ± 1.6 nm) than those originating from the other two cell lines (*p* < 0.001) (Figure 3D), which measured 162.5 ± 5.9 nm and 17.3 ± 7.3 nm in PEO1 cell and OAW42 cells, respectively. We therefore took a closer look at the distribution of sizes represented by so-called percentile values. The D10/D50/D90 values were as follows: OAW28-derived EVs: 123.8/206.7/379.7 nm; PEO1-derived EVs: 101.6/139.9/259 nm; OAW42-derived EVs: 79.5/133.3/251 nm. All percentile values of EVs derived from OAW28 cells were significantly (*p* < 0.001) higher than those of PEO1-derived EVs and OAW42-derived EVs (Figure 3D). The D50 value (206.7 ± 7.2 nm) of OAW28-derived EVs showed that approximately 50% of released EVs were medium/large in size (>200 nm) (Figure 3D), which was also confirmed by TEM (Figure 2A). PEO1 and OAW42 released fewer medium/large (>200 nm) EVs: 23% and 15%, respectively (Figure 3B,C). On the other hand, PEO1-derived EVs and OAW42-derived EVs did not significantly differ in D50 and D90 values. However, a higher proportion of EVs below 100 nm was revealed by the D10 value of OAW42-derived EVs (79.5 ± 8.4 nm), which was statistically significantly smaller than with OAW28-derived EVs (139.9 ± 3.9 nm; *p* < 0.001) and PEO1-derived EVs (133.3 ± 6.4 nm; *p* < 0.02) (Figure 3D).

### 2.2. EV Concentration: The Determination of the Level of EVs Released from OC Cells from the Clinically Confirmed Chemoresistant Patient Is Possibly Not Enough to Predict the Response to Platinum-Based Therapy

To determine the EV concentrations, the same samples were analyzed by two methods, S-NTA and FT-FCM (Figure 4A,B). As already noted, we refer to ‘‘particles’’ rather than ‘‘vesicles’’ in the context of S-NTA analysis. The mean concentration of particles detected by S-NTA for OAW28 cells was 6.12 ± 0.36 × 10^8^/mL per 10^6^ cells, and the particle concentrations of PEO1 and OAW cells were 5.43 ± 0.08 × 10^8^/mL and 2.48 ± 0.15 × 10^8^/mL per 10^6^ cells, respectively (Figure 4A). For the FT-FCM analysis, we labeled the EVs with calcein. Calcein is activated by an enzyme within an EV, so its advantage is the differentiation of intact EVs from membrane fragments [33,34]. The results of FT-FCM showed that OAW28 cells produced 2.11 ± 0.07 × 10^8^/mL calcein-positive EVs per 10^6^ cells, whereas PEO1 and OAW42 cells produced 2.27 ± 0.03 × 10^8^/mL and 0.7 ± 0.02 × 10^8^/mL calcein-positive EVs per 10^6^ cells, respectively, in the same time period (Figure 4B). According to the results of both methods, the number of OAW28-derived calcein-positive EVs detected by FT-FCM and OAW28-derived particles detected by S-NTA did not significantly differ from those of PEO1 cells. However, the number of calcein-positive EVs/particles released from OAW42 cells was significantly lower (*p* < 0.001) than the number of calcein-positive EVs/particles released from OAW28 and PEO1 cells (Figure 4A,B). Despite the lower particle number detected by FT-FCM, the two methods, FT-FCM and S-NTA, produced results with high concordance (*r* = 0.95, *p* < 0.001).

### 2.3. EpCAM-Specific EVs: A Promising Biomarker to Predict Chemoresistance to Platinum-Based Therapy

EpCAM-positive vesicles were detected by FT-FCM (Figure 5A). The mean concentration of EpCAM-positive EVs for OAW28 cells was 9.19 ± 0.32 × 10^7^/mL per 10^6^ cells, while for PEO1 and OAW cells the concentrations were 1.78 ± 0.16 × 10^7^/mL and 0.25 ± 0.005 × 10^7^/mL per 10^6^ cells, respectively, released in the same time period (Figure 5A left axis). The mean concentration of EpCAM-positive EVs released from OAW28 cells was significantly higher (*p* < 0.001) than the mean concentration released from PEO1 cells and from OAW42 cells. In addition, the number of EpCAM-positive EVs released from PEO1 was significantly higher (*p* = 0.002) than the number of EVs released by OAW42. 

To confirm the results of FT-FCM analysis, EVs were also specifically isolated using anti-EpCAM magnetic beads and analyzed by BB-FCM (Figure 5A right axis). The results obtained by this method confirmed that the number of EpCAM-positive EVs released from OAW28 (80.23 ± 2.72% EpCAM-positive beads) was significantly higher (*p* < 0.001) than the number of EpCAM-positive EVs released from PEO1 (37.34 ± 1.2% EpCAM-positive beads) or OAW42 (17.78 ± 1.45% EpCAM-positive beads) cells. Even so, the concentration of PEO1-derived EVs was significantly higher (*p* < 0.001) than those released from OAW42 cells.

The presence of EpCAM on the membrane of EVs was also confirmed by TEM. TEM showed the presence of EpCAM in the different size ranges of larger and smaller EVs (Figure 5B). However, immunogold labeling predominated in EVs that were ≥400 nm in size. A higher number of EpCAM molecules was found on larger EVs (Figure 5B).

If EpCAM-positive EVs are generated by budding outward from the plasma membrane, then the higher expression of EpCAM on the plasma membrane should generate a higher number of EpCAM-positive EVs. To ascertain whether this positive correlation exists, we determined the EpCAM expression on the plasma membrane by FCM (Figure 6A,B). The expression of EpCAM on the cell membrane was significantly higher in the OAW28 cell line than in the PEO1 (*p* = 0.004) and OAW42 (*p* < 0.001) cell lines (Figure 6A). EpCAM expression was also significantly higher in the PEO1 cell line than in the OAW42 cell line (*p* = 0.009). We next compared the expression of EpCAM on the cell membrane with the concentration of EpCAM-positive EVs. The cellular protein and the concentration of EpCAM-positive EVs showed a strong positive correlation (*r* = 0.950, *p* < 0.001; Figure 6B).

## 3. Discussion

Despite novel targeted drugs, platinum-resistant OC remains a fatal disease. Moreover, there are currently no validated molecular predictive biomarkers for platinum resistance. The identification of new predictive biomarkers to circumvent platinum resistance is thus highly desirable [4]. Tumor-derived EVs are a promising source of such biomarkers, especially since their cargo, concentration and size have been shown to be unique in tumor states [36,37,38,39,40]. Publications on EVs in the field of OC chemoresistance are primarily dedicated to the research of EV-mediated drug-resistance mechanisms [40,41]. The main proposed roles of EVs in this field are drug export, antigen sink, the transfer of the resistant phenotype and alternation in the tumor microenvironment [38,40]. However, the potential of EVs for predicting responses to platinum-based therapy in OC remains largely unexplored.

In this study, three OC cell lines, OAW28, OAW24 and PEO1, were used as the EV source in order to investigate the characteristics of EVs from a patient who did not respond to cisplatin (OAW28) compared to two patients with a complete response (PEO1 and OAW42) [42,43]. Since EVs are a promising tool for liquid biopsy, we chose cell lines that had been isolated from patients’ ascites [42,43]. We ensured that all three cell lines were well characterized. The OAW28 cell line is among the twelve best candidates for in vitro studies of HGSOC. The evaluation was based on the correlation of genomic characteristics between tumor samples and cell lines. For this purpose, Domcke et.al. performed a data comparison between The Cancer Genome Atlas, (TCGA) and The Broad–Novartis Cancer Cell Line Encyclopedia (CCLE) [33]. In addition, among the 39 OC cell lines tested, the highest concentration of carboplatin causing 50% growth inhibition was determined for the OAW28 cell line [34]. In relation to new strategies in the therapy of HGSOC, it is also important that OAW28 does not have *BRCA1/2* mutations [33,34]. For a direct comparison of HGSOC EV characteristics, we selected another HGSOC cell line, PEO1 [34,35]. Interestingly, the PEO1 cell line exhibits a reversion of the deleterious *BRCA2* mutation, indicating selective pressure against *BRCA1/2* mutations. Additionally, a loss of heterozygosity (LOH) was detected at both *BRCA1* and *BRCA2* loci [44]. The third cell line, OAW42, is probably not part of the HGSOC subtype, although the original annotation was serous [33,34,43]. However, during culturing, OAW42 cells show ‘epithelial’ morphological characteristics [34]. We chose this cell line in order to determine whether differences in EV characteristics could be the result of different tumor subtypes. OAW42 shows no *TP53* mutation but a *PIK3CA* mutation [33,34]. The examination of the *BRCA1* status yielded contradictory results; a *BRCA1* mutation was detected in one study, but this result was not confirmed in two other studies [33,34,44]. In addition, OAW42 showed no LOH at the *BRCA1/2* locus [44].

Cells release a heterogeneous population of EVs with different sizes (30–10,000 nm) [45]. EVs are a mixture of exosomes and ectosomes (microparticles/microvesicles), which differ in size and biogenesis. Exosomes are released in the process of the exocytosis of multivesicular bodies (MVBs). Ectosomes, on the other hand, bud directly outwards from the plasma membrane and tend to be larger in size. Due to overlapping size ranges, current EV isolation methods are not able to differentiate precisely between subtypes of EVs, so assigning an EV to a particular biogenesis pathway remains difficult. We therefore refer to “small EVs” (sEVs) and “medium/large EVs” (m/lEVs) based on the size range: <200 nm is small and >200 nm is large and medium, as recommended by the International Society for Extracellular Vesicles [46]. The TEM analysis revealed that all three cell lines released spherical EVs with different sizes, although a more heterogeneous size distribution and a higher proportion of m/l EVs was observed among EVs released from OAW28 cells, which was confirmed by the results of S-NTA. On the other hand, the EVs from all three cell lines were most commonly found in the size range of 100 to 140 nm, which falls within the size range of sEVs. This result is comparable with a previously published study performed with an OVCAR3 cell line, as another cell line of the HGSOC subtype, and analyzed with S-NTA [18,34]. However, the most frequently found size of EVs released from OAW28 cells was larger than that from PEO1-derived EVs, while OAW42-derived and OAW28-derived EVs did not differ. This observation indicates that the size of the major peak of EVs depends on other factors, such as a different histological subtype [18]. On the other hand, a higher proportion of m/l EVs released from OAW28 was observed compared to PEO1-derived and OAW42-derived EVs. This result is therefore probably not due to differences between EVs released from different histological subtypes but may indicate an association of a higher number of m/l EVs with resistance to chemotherapy. The difference in the size distribution of EVs is probably a very early sign of an altered state of the cell, which is reflected in the biogenesis of EVs. It has been observed that the proportion of EVs of different sizes released from HeLa cells was modified due to the silencing of some cell components involved in EV formation [36]. One problem in determining the significance of the modified size distribution is that its analysis also depends on the method of EV isolation. Although the size distribution determined with S-NTA showed a major EV peak independently of the isolation method, we still do not know which subpopulations are lost during the isolation process [45]. This problem is even greater when EVs are isolated from a patient’s body fluid. Although studies on the association of a larger size of EVs with a worse prognosis already exist for EVs obtained from the blood of patients with pancreatic ductal adenocarcinoma [47,48], such research is still lacking in the field of HGOSC. Despite the problems in analyzing EVs, tumor-derived large EVs have progressively received increasing interest due to their peculiar content and function [49]. 

Since the quantification of EVs is difficult, we used two methods to confirm our results: S-NTA and FT-FCM. Particle counting with S-NTA may result in an overestimation of EV counts since the technique is not specific to EVs and also registers co-isolated particles. On the other hand, FT-FCM, due to the detection limit, registers larger particles than S-NTA. We used calcein as a generic fluorescent marker for detecting EVs by FT-FCM. The advantage of calcein is that it distinguishes intact EVs from membrane fragments, while the disadvantage is that it requires the presence of enzymes in the EVs. For both reasons, particle counting with FT-FCM may result in an underestimation of EV counts. However, the results of S-NTA and FT-FCM were in excellent correlation (*r* = 0.95, *p* < 0.001) despite the lower particle number detected by FT-FCM, which is a logical consequence of the above explanation.

In the present study, in which the results were obtained using the two methods mentioned above, we demonstrated that the EV concentration between OAW28-derived EVs and PEO1-derived EVs did not differ significantly, which indicates that HGSOC cells from patients resistant to platinum-based therapy do not release more EVs than cells from tumors sensitive to platinum-based therapy. The detected concentration of EVs from OAW28 and PEO1 cell lines analyzed by S-NTA in our study were in accordance with previous reports on the EV concentration released from another HGSOC cell line (OVCAR3) in the same period of time and analyzed using the same method [18,34]. Intriguingly, the concentration of EVs produced by OAW42 cells was lower than the concentration of EVs produced by OAW28 and PEO1 cells. This difference in the production of EVs between two cell lines from tumors sensitive to platinum-based therapy might be attributable to different histological OC subtypes. However, the detected EVs released from OAW42 were much smaller than the results of a previous study. Zhang et al. measured the concentration of EVs released from cell lines with HGS histology, endometrioid histology and clear cell histology, but the number of EVs produced by cells from different histological OC subtypes did not differ [18]. The lower EV concentration produced by the OAW42 cell line might therefore be a consequence of its particular genomic profile or some other unknown factor [33,34]. Szajnik et al. investigated the relationship between plasma EV level (measured as protein level) and responses to primary therapy with carboplatin/paclitaxel in 12 OC patients with different types of histology. The levels of EV proteins varied widely among patients, but the response to therapy was not associated with the level of plasma EV proteins before the initiation of the treatment, which is in agreement with the results of our study. However, the clinical response to therapy in the majority of patients was associated with a decrease in EV protein levels. These results, while preliminary and obtained with only a few patients, should be further evaluated [17].

It is likely that specific EVs might be more informative in predicting the response to therapy than total EV levels. Among specific cargos of EVs, we chose EpCAM as a marker of epithelium-derived cells. EpCAM-positive vesicles were detected in the plasma, ascites and pleural effusion of OC patients, as well as in conditioned media of OC cell lines [7,8,15,18,19]. However, the relevance of this analysis of EpCAM-positive EVs in predicting chemoresistance has been largely unexplored. The results of our study, obtained by two methods, FT-FCM and BB-FCM, showed that cells derived from the patient resistant to platinum-based therapy (OAW28) released a much larger number of EpCAM-positive EVs than cells from tumors sensitive to platinum-based therapy, regardless of the histological subtype. EpCAM-positive vesicles released from OAW28 cells accounted for 43% of total EVs detected by FT-FCM. A previous study reported a lower percentage of EpCAM-positive EVs released from another HGSOC cell line (~24 %, OVCAR-3), an endometrioid cell line (~18%, ES-2) and a clear cell carcinoma cell line (~15%, IGROV) detected by fluorescent-NTA [18,34].

EpCAM-positive vesicles showed the clinical usefulness of monitoring the clinical response to platinum-based therapy. The EVs’ EpCAM levels decreased among responding patients, whereas increased levels were associated with non-responding patients, as detected by a nano-plasmonic sensor. However, the difference was not significant since the study was performed on eight OC patients with different types of histology. The levels of the EVs’ EpCAM before therapy varied greatly among patients, but the relation of the response to therapy with these levels was not assessed [8]. EpCAM was also included in the EV signature (a weight sum of eight EV protein markers), which provides a complementary approach for monitoring treatment responses in metastatic breast cancer [40].

These studies and results raise an interesting question: what is the content of EpCAM-positive EVs? Taylor et.al. isolated EpCAM-positive EVs from the sera of 50 patients with HGSOC and detected eight microRNA (miR-21, miR-141, miR-200a, miR-200b, mir200c, miR-203, miR-205 and miR-214) in EVs for which levels were increased compared to those with benign diseases [7]. Although the purpose of this study was to discover diagnostic biomarkers, there are other published studies that have examined the presence of noncoding RNA (microRNA and circular RNA) in EVs and their utility as biomarkers of responses to chemotherapy [40,50,51,52]. Moreover, in prostate cancer, miR-200c and miR-205 have been shown to induce the expression of EpCAM mRNA and protein [53]. Interestingly, a higher level of miR-21 in OC exosomes was associated with paclitaxel resistance [54].

In the CTC analysis, EpCAM antigen is used to obtain EpCAM-positive cells from blood, alone or in combination with other antigens [23,30]. Chebouti et al. detected CTCs using immunomagnetic CTC enrichment targeting EpCAM, MUC-1 and CA-125, followed by RT-PCR to detect the expression of EpCAM, MUC-1, CA-125 and excision repair cross-complementary group 1 (ERCC1) protein in isolated CTCs. The assessment of ERCC1-expressing CTCs alone had prognostic relevance, while positive results for at least one additional marker (EpCAM, MUC-1 or CA-125) increased the level of statistical significance of the result of ERCC1 alone. Although CTCs were captured by a procedure that targets EpCAM and MUC-1 surface epitopes, EpCAM and MUC-1 mRNA seemed to be downregulated. The authors explained that the discordance between the proteins and transcript expression profiles of the cell could be due to the post-transcriptional modification of mRNA or differences in the half-life between mRNAs and their corresponding proteins [22]. A direct comparison of EpCAM proteins and EpCAM RNA levels between CTCs and EVs preferentially from the same sample might explain this discrepancy. One of the *pros* of EVs in comparison to CTCs is that RNA, DNA and proteins are protected by the EV membrane from nucleases, proteases, fluctuations in pH and other environmental factors [55,56]. There has recently been an intensification of research examining the value of complementary liquid biopsies [56]. Moreover, we already have the results of a study in metastatic breast cancer, suggesting that tumor-derived EVs in association with CTCs may help in selecting more effective therapies [57]. 

We found that the cells from the chemoresistant patient released EpCAM-positive EVs of different sizes (sEVs and m/lEVs). However, a higher proportion of m/l EVs was observed compared to those from cells from the tumors sensitive to chemotherapy. Additionally, immunogold labeling predominated in EVs that were ≥400 nm in size, and a higher number of EpCAM molecules was found on larger EVs. Gercel-Taylor et al. also observed the presence of EpCAM in various size ranges of vesicles derived from the sera of advanced HGSOC patients [14]. 

We hypothesized that larger EpCAM-positive EVs with a higher number of EpCAM molecules can bind more effectively to tumor-reactive autoantibodies and to therapeutic EpCAM antibodies and thereby reduce the binding of antibodies to tumor cells. It has been discovered that the ectopic overexpression of EpCAM and HER2 in cancer cells induces humoral immune responses in patients. Both antigens are also targets for a number of therapeutic monoclonal antibodies. In the first situation, this leads to the impairment of antibody-dependent cellular cytotoxicity (ADDC), which is a major immune effector function toward tumor cells [58]. EpCAM-specific autoantibodies have been found in the sera of OC patients [59]. This phenomenon may be an additional explanation of the contradictory results of EpCAM overexpression in ovarian tumors in relation to its prognostic value [20,21,25,26]. In the second situation, this may contribute to therapeutic failure of EpCAM monoclonal antibodies. This has been suggested in the case of the HER-2-specific therapeutic antibody, trastuzumab [58]. A higher level of HER2-specific autoantibodies was found in sera from patients with HER2-expressing mammary carcinoma [60]. It has been suggested that the antibody has a higher binding affinity to full-sized HER2 than to soluble HER2 extracellular domains generated by proteolytic cleavage [61]. It would therefore be worth exploring the role of this indirect mechanism of EpCAM on EVs in therapeutic failure.

The results of the present study showed a strong positive correlation between the concentration of EpCAM-positive EVs and the expression of EpCAM on the cell membrane. This finding supports the use of EpCAM-specific EVs as cellular surrogates for EpCAM and could therefore be included in liquid biopsies to shed light on the contradictory results of the immunohistochemical evaluation of EpCAM overexpression in ovarian tumors as a biomarker for predicting responses to chemotherapy. Moreover, it would be interesting to examine in a future study the clinical usefulness of EpCAM-specific EVs in combination with other promising molecular biomarkers or other innovative approaches for the prediction of platinum-resistant HGSOC, especially EVs in combination with the detection of CTCs and cell-free DNA.

In conclusion, we showed for the first time that a more heterogeneous size distribution, with a higher proportion of larger EVs, is characteristic of EVs released from the HGSOC cell line of a clinically confirmed resistant patient, which may indicate the changed state of the cell and is reflected in the biogenesis of EVs. In addition, these cells released a higher number of EpCAM-specific EVs of different sizes, although the expression of EpCAM predominated on larger EVs, especially on those larger than 400 nm. The association of a higher concentration of EpCAM-specific EVs with a higher expression of EpCAM on cell membranes demonstrates the feasibility of applying EpCAM-specific EVs as a biomarker for cellular EpCAM. Although the results of the current study should be validated in clinical samples, they reveal novel characteristics of EVs that may have the potential to aid the prediction of platinum-resistant HGSOC and thereby enable the selection of personalized therapy.

## 4. Materials and Methods

### 4.1. Cell Culture

We included in this study three well-characterized OC cell lines purchased from the European Collection of Authenticated Cell Cultures (ECACC via Sigma): OAW28 (ECACC 85101601), PEO1 (ECACC 10032308) and OAW42 (ECACC 85073102). In vitro cultured OAW28, OAW24 and PEO1 cells were used as the EV source for this study in order to examine the characteristics of EVs from a patient who did not respond to cisplatin (OAW28) compared to EVs from two patients with a complete response (PEO1 and OAW42) [42,43]. OAW28 and PEO1 are representatives of HGSOC, while OAW42 cells were described to be of serous origin without mutation in the *TP53* gene [33,34]. Cell lines were cultured according to the manufacturer’s instructions. OAW28 and OAW42 cells were grown in DMEM medium (Sigma-Aldrich, St. Louis, USA; #D6546) and supplemented with 2 mM glutamine, 20 UI/l bovine insulin purchased from Sigma-Aldrich, 1 mM sodium pyruvate, 10 % FBS (Gibco/Thermo Fischer Scientific Inc., Waltham, MA, USA; #10500-037, Lot 08G1094K), 100 U/mL penicillin, 100 µg/mL streptomycin and 50 µg/mL gentamycin purchased from Gibco/Invitrogene. PEO1 cells were grown in RPMI 1640 medium (Gibco, Paisley, UK; #A10491-01), supplemented with 1 mM sodium pyruvate, 10 % FBS (#10500-037, Lot 08G1094K), 100 U/mL penicillin, 100 µg/mL streptomycin and 50 µg/mL gentamycin purchased from Gibco/Invitrogene. The cell lines were maintained at 37 °C in an incubator with humidified air with 5% CO_2_. For EV isolation, cells were seeded at a density of 1.5 × 10^6^, grown in a 175 cm^2^ flask until ~80% confluence, rinsed three times with Dulbecco’s phosphate-buffered saline (DPBS, without Ca^2+^ and Mg^2+^; Gibco/Thermo Fischer Scientific Inc., #14190-94) and grown in 30 mL serum-free medium. After 24 h of incubation, the conditioned medium was collected and centrifuged for 30 min at 2000× *g* RT to eliminate cell and cell debris contamination. Supernatant 10 mm above the pellet was transferred into a plastic tube and mixed gently before proceeding with isolation of EVs. For analysis of expression of EpCAM on the cell membrane and for cell/viability counts, attached cells were fully disaggregated by TripLE^TM^ select enzyme (Gibco/Thermo Fischer Scientific Inc.; #12563-029). Cell lines were used from passages 4–7.

### 4.2. Cell Counts and Viability Measurement

To normalize the amount of released EVs, we chose a number of live cells [34]. For viability measurement, the cell suspension was mixed with an equal volume of 0.4% trypan blue solution (Invitrogen/Thermo Fischer Scientific Inc., Charlotte, USA; #T102882) to count live and dead cells using an image-based automated cell counter (Countess II FL; Thermo Fisher Scientific Inc., Waltham, MA, USA) The required viability was ≥90%.

### 4.3. EV Enrichment (Isolation)

The cell culture conditioned medium sample (7 mL) was transferred to a fresh tube, and 3.5 mL of exosome isolation reagent (Invitrogen/Thermo Fischer Scientific Inc., #4478359) was added and mixed thoroughly by pipetting up and down. The solution was incubated overnight at 4 °C. The following day, the samples were centrifuged at 10,000× *g* for 60 min. The pellet was then resuspended in 200 µL DPBS and stored at 2–8 °C for a maximum of up to seven days before downstream analysis or further purification by the immunomagnetic bead method.

### 4.4. Transmission Electron Microscopy (TEM)

Isolated EVs were fixed with 2% paraformaldehyde (in PBS) and 10 µL drop of EVs was loaded onto a glow-discharged formvar/carbon-coated grid. Adsorbed vesicles were washed in distilled water, stained by negative contrast with 1% aqueous uranyl acetate, briefly washed in distilled water and air-dried. Samples were examined with a Philips CM100 transmission electron microscope operated at 80 kV. Images were captured with an AMT camera (Advanced Microscopy Techniques Corp., Woburn, MA, USA).

For EpCAM immunolabeling, a 10 µL drop of paraformaldehyde-fixed EVs was loaded onto a glow-discharged formvar/carbon-coated grid. Adsorbed vesicles were washed in distilled water, blocked in 0.5% BSA, incubated with mouse monoclonal anti-EpCAM (CD326) primary antibodies, clone 1B7 (Invitrogen/Thermo Fisher Scientific, Waltham, MA, USA; #12-9326-42 diluted 1:100 in 1% BSA/PBS) for 30 min at room temperature. Vesicles were washed in PBS and incubated with 18 nm colloidal gold goat anti-mouse IgG (Jackson ImmunoResearch, Cambridgeshire, CB7 4EX UK; #115-215-146, diluted 1:40 in 1% BSA/PBS) secondary antibodies, washed in distilled water and stained with 1% aqueous uranyl acetate. Air-dried samples were examined with a Philips CM100 transmission electron microscope operated at 80 kV and equipped with an AMT camera.

### 4.5. Nanoparticle Tracking Analysis in Scatter Mode (S-NTA)

Isolated EVs from conditioned media were diluted in filtered DPBS (without Ca^2+^ and Mg^2+^) and analyzed for particle size distribution and concentration by S-NTA using a NanoSight NS300 instrument (488 nm laser) connected to an automated sample assistant (both Malvern Panalytical, Worcestershire, UK). The samples were diluted to a concentration allowing accurate particle tracking (according to the manufacturer, app. 20–100 particles in the field of view). Each sample was measured (recorded) four times with 80 s acquisitions at 25 °C and processed using NTA software (version 3.3). The camera level was adjusted according to the sample. Analysis settings were maintained for all readings: detection threshold 5, water viscosity, automatic blur size and automatic (10.2–15.7 pix) maximum jump distance. Data obtained were particle concentration (number of particles/mL of cell-cultured conditioned media), mode (nm), mean (nm) and percentile values: D10, D50 and D90. Percentile values indicated that 10, 50 and 90%, respectively, of the particle distribution was below this value. To calculate the number of particles for each sample, the dilution factor during sample preparation was taken into account. Reference nanospheres (100 nm polystyrene, Malvern Pananalytical, Worcestershite, UK; #LT3100A) were analyzed on the same day to compensate for day-to-day variation in instrument performance. Raw data were analyzed by NanoSight NTA 3.3 software (Malvern Panalytical, Worcestershire, UK).

### 4.6. Fluorescence-Triggered Flow Cytometry (FT-FCM) of EVs

A flow cytometer (CytoFLEX, Beckman Coulter, Brea, CA, USA) was used to determine the concentration of EpCAM-positive and calcein-positive EVs in the conditioned media of OC cell lines. Fluorescence-based detection of EVs using FCM was employed to better resolve EVs from the background noise [62]. Calcein-acetoxymethyl ester (AM) green (referred to as calcein) (Thermo Fischer Scientific; #C3100MP) was used as the generic fluorescent marker for the detection of EVs. Calcein AM is non-fluorescent until it passively enters EVs, after which the lipophilic blocking groups are cleaved by non-specific esterase. As a result, calcein is trapped inside the EVs and emits a green fluorescent signal (emission max. 516 nm) following excitation with a blue (488 nm) laser [46,63,64]. For sample preparation, antibodies were centrifuged at 20,000× *g* for 30 min at RT to remove any fluorescent antibody aggregates, as described elsewhere [65]. To stain, 20 µL of sample was incubated with 1.25 µL PE-conjugated anti-EpCAM (CD326) primary antibodies, clone 1B7 (Invitrogen/ Thermo Fisher Scientific, Waltham, ZDA; #12-9326-42), or PE-conjugated isotype antibody, mouse IgG1 kappa (Invitrogen/Thermo Fisher Scientific, Waltham, ZDA; #12-1714-42), and kept in the dark for 2 h at room temperature. Due to some overlap between the emission spectra of PE and calcein, which could lead to a false positive signal, a separate sample was stained with calcein [66]. A total of 100 µL of calcein (10 µM) was added to 20 µL of sample and incubated for 30 min in the dark on ice. After incubation, we diluted samples in DPBS to a count rate below 2000 events/s to prevent swarm detection [67].

The stained samples were measured by flow cytometry within 1.5 h, using fluorescence triggering, and timed at a slow flow rate for 120 s. Enumeration of EVs was performed using volumetric measurement (events/mL). Calibrating the sample flow rate was conducted following the CytoFLEX instructions by water weight difference during 18 min acquisition with a slow flow rate. To set up VSSC detection, we replaced the 450 nm filter with a 405 nm filter and appropriately labeled the detector as VSSC. For daily verification of the flow cytometer’s optical alignment and fluidics system, we used CytoFLEX Daily QC Fluorospheres (Beckman Coulter, Brea, CA, USA; #B53230) with settings optimized for EV detection. The upper size limit for the EV gate was set by detecting 1000 nm fluorescent-green silica beads (Kisker-biotech, Steinfurt, Germany; #PSI-G1.0) using VSSC triggering [56]. The threshold for fluorescence-triggering of calcein-positive events was set by the plain cell culture medium samples, which were treated with calcein in the same way as the EVs in the culture conditioned medium samples. The fluorescence signal from PE was used to trigger detection of EVs labeled with anti PE-conjugated antibodies. The gate for PE was derived by measuring the corresponding isotype control bodies. Another negative control was performed using anti-EpCAM-PE in DPB and unstained EV samples. The addition of 0.2% Triton 100 (Sigma, St. Louis, USA; #T8787) to EV samples for 20 min at room temperature was accompanied by a near total disappearance of the EVs. Analysis of the acquired data was performed using CytExpert 2.3 software (Beckman Coulter, Brea, CA, USA). To calculate the number of events for each sample, the dilution factor during sample preparation and staining was taken into account. Due to the detection limit of our flow cytometer, the EVs measured in this study predominantly correspond to ectosomes (microparticles) and not to exosomes. 

### 4.7. Anti-Human EpCAM Antibody-Coated Magnetic Bead-Based EV Isolation

EpCAM-positive EV subsets from a pre-enriched EV sample were isolated using Exosome-Human EpCAM Flow Detection (Invitrogen/Thermo Fischer Scientific Inc., #10624D) following the manufacturer’s instructions. Dynabeads^TM^ are magnetic polystyrene beads (2.7 µm diameter) coated with primary monoclonal antibodies specific for the EpCAM membrane antigen expressed on the EVs. The assay buffer was prepared from DPBS with added 0.1% BSA as the blocking agent, and passed through a 0.2 µm filter Dynabeads were resuspended by vortexing for 30 s. Then, 20 µL was withdrawn and transferred to a tube with 1 mL of assay buffer. The tube was placed in a DynaMag™-2 magnetic separator (Thermo Fischer Scientific Inc., #12321D) for 2 min. After the buffer had been carefully removed, 90 µL of fresh assay buffer and 10 µL of pre-enriched EV sample were added and incubated at 2–8 °C overnight on a HulaMixer (Thermo Fischer Scientific Inc., #15920D). The next day, the assay buffer was added, and the tube was placed in a magnetic separator for 1–2 min before removing all supernatants. The procedure with the magnetic separator was repeated 3 times to wash the bead-bound EVs. For analysis with bead-based flow cytometry, the sample was finally diluted in 300 µL of assay buffer. 

### 4.8. Bead-Based Flow Cytometry (BB-FCM) of EpCAM-Positive EVs

EVs were also analyzed by bead-based flow cytometry (BB-FCM) as a semi quantitative method for assessing approximate EpCAM-positive concentrations [68]. Because of the size of the Dynabeads™ magnetic beads, EVs are easily detected by flow cytometry while bound to the surface of the beads, enabling the detection of EpCAM on the EV membrane. A total of 100 µL of bead-bound EV sample was stained using 20 µL (0.06 µg) of RPE-conjugated anti-EpCAM (CD326) primary antibodies, clone EBA (Biosciences/Becton Dickinson, San Jose, CA, USA; #347198). After 45 min of incubation in the dark on an orbital shaker (RT, 1000 rpm), 300 µL of assay buffer was added to each tube, and the tubes were placed in a magnetic separator for 1–2 min before removing the buffer. We repeated the procedure with the magnetic separator once more. The final EV sample for flow cytometry was diluted in 300 µL DPBS. Flow cytometry was performed using a CytoFLEX flow cytometer (Beckman Coulter, Brea, CA, USA). Isotype control antibodies (Mouse IgG1-RPE isotype control, Biosciences/Becton Dickinson; #559320) and an unstained EV sample were used as negative control. Gating of bead-bound EVs was performed based on FSC/SSC parameters so that non-bead events and bead doublets were excluded from the analysis, and beads were selected using the corresponding channels of fluorescence. Results were expressed as EpCAM-positive beads. Analysis of the acquired data was performed using CytExpert 2.3 software (Beckman Coulter, Brea, CA, USA).

### 4.9. Flow Cytometry of EpCAM Cell Membrane Expression

Expression of EpCAM on the cell membrane was analyzed using a CytoFLEX flow cytometer (Beckman Coulter, Brea, CA, USA). For analysis with the flow cytometer, 2 × 10^5^ cells were needed for each sample, which were resuspended in 100 µL of DPBS. A total of 1.25 µL PE-conjugated anti-EpCAM (CD326) primary antibodies, clone 1B7 (Invitrogen/Thermo Fisher Scientific, Waltham, ZDA; #12-9326-42), were added to the first sample. The second sample was an isotype control, so 1.25 µL of PE-conjugated isotype antibody, mouse IgG1 kappa (Invitrogen/Thermo Fisher Scientific, Waltham, ZDA; #12-1714-42), was added to the cells. The third sample consisted of unstained cells. The samples were incubated in the dark for 45 min at 4 °C. They were then washed three times by centrifugation (3 min, 200× *g*, at room temperature) and resuspended in between in 500 µL DPBS. Final cell samples for analysis were resuspended in 200 µL DPBS. Fluorescence gating parameters were established using antibody isotype controls, and values above a 99% negative staining threshold were considered positive. Using isotype antibodies, we confirmed that the binding of primary antibodies was specific. Unstained cells were used to control cellular autofluorescence. The concentration of the isotype antibodies should be the same as the concentration of the primary antibodies. We set up the instrument appropriately with the help of this control, which also allowed us to set up the gate appropriately to capture the events we wanted to analyze. A total of 5000 cells were measured. Mean fluorescent intensity (MFI) was used to compare the expression of EpCAM between different cell lines. Analysis of the acquired data was performed using CytExpert 2.3 software (Beckman Coulter, Brea, CA, USA).

### 4.10. Statistical Analysis

Statistical analysis was performed using SigmaPlot v.12.5 (Systat Software, Palo Alto, CA, USA). All quantitative measurements were conducted as three independent measurements, which were made in duplicate. One-way ANOVA was used as an appropriate method to test for significant differences between more than two normally distributed samples, followed by Holm–Sidak’s multiple comparison test. Pearson’s correlation coefficient was used to calculate the direction and strength of the relationship between variables. A *p* value of <0.05 was considered significant. All data are presented as mean ± standard error of mean (SEM).

## Figures and Tables

**Figure 1 pharmaceuticals-16-00907-f001:**
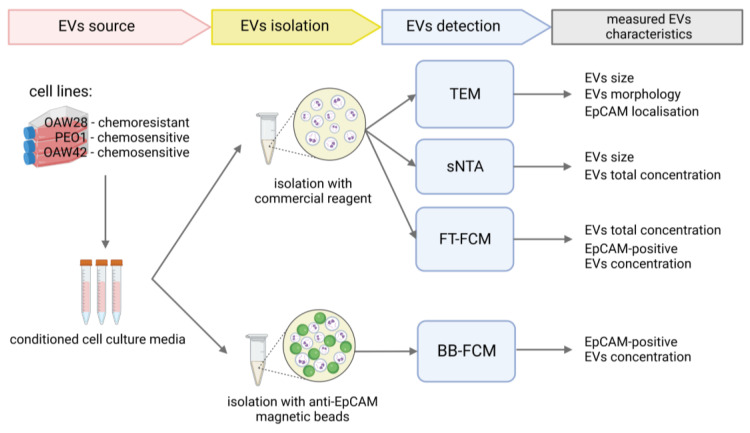
Graphic presentation of the methods.

**Figure 2 pharmaceuticals-16-00907-f002:**
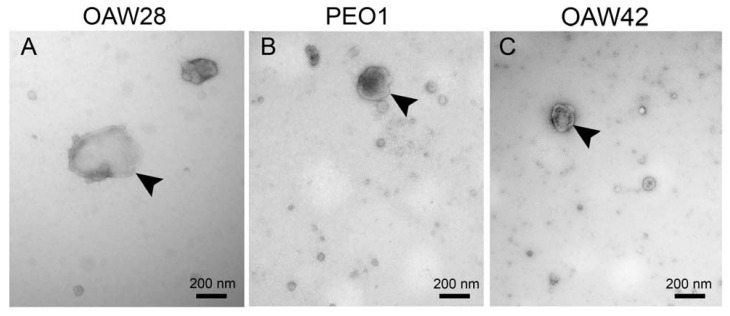
Transmission electron microscopy analysis of extracellular vesicles (EVs) released from OAW28 (**A**), PEO1 (**B**) and OAW42 (**C**) cells. Negative contrast staining shows the typical vesicular morphology of EVs. In the isolate of OAW28, the EVs are larger ((**A**), arrowhead) than the EVs of PEO1 ((**B**), arrowhead) and OAW42 ((**C**), arrowhead) cells.

**Figure 3 pharmaceuticals-16-00907-f003:**
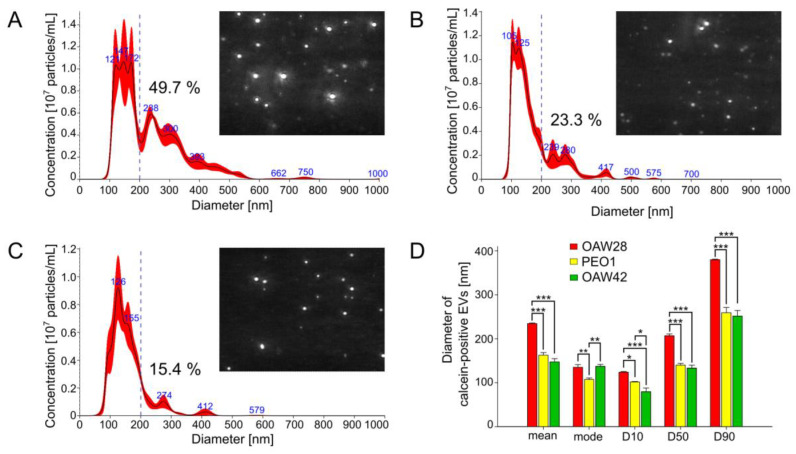
Nanoparticle tracking analysis with representative live stream views of extracellular vesicles (EVs) released from OAW28 (**A**), PEO1 (**B**) and OAW42 (**C**) cells. OAW28 cells released more medium/large (>200 nm) EVs (**A**) than PEO1 (**B**) and OAW42 (**C**) cells. The bar graph shows a comparison of mode, mean, D10, D50 and D90 diameters among all three cell lines. Particle size distribution D10, D50 and D90 corresponding to the percentages 10%, 50% and 90% of particles under the reported particle size (**D**). Data are presented as mean ± SEM of three independent measurements made in duplicate. * *p* < 0.05; ** *p* < 0.01; *** *p* < 0.001.

**Figure 4 pharmaceuticals-16-00907-f004:**
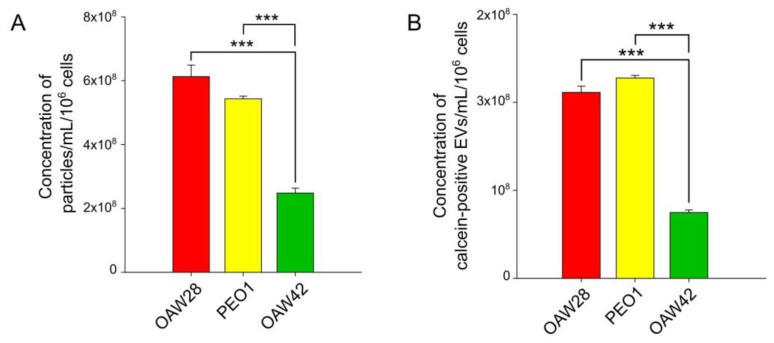
Concentration of total extracellular vesicles (EVs) (detected as particles/calcein-positive EVs) released in 24 h from OAW28 (red bar), PEO1 (yellow bar) and OAW42 (green bar) cells analyzed by nanoparticle tracking analysis (**A**) and fluorescence-triggered flow cytometry (**B**). Data are presented as mean ± SEM of three independent measurements made in duplicate. **** p* < 0.001.

**Figure 5 pharmaceuticals-16-00907-f005:**
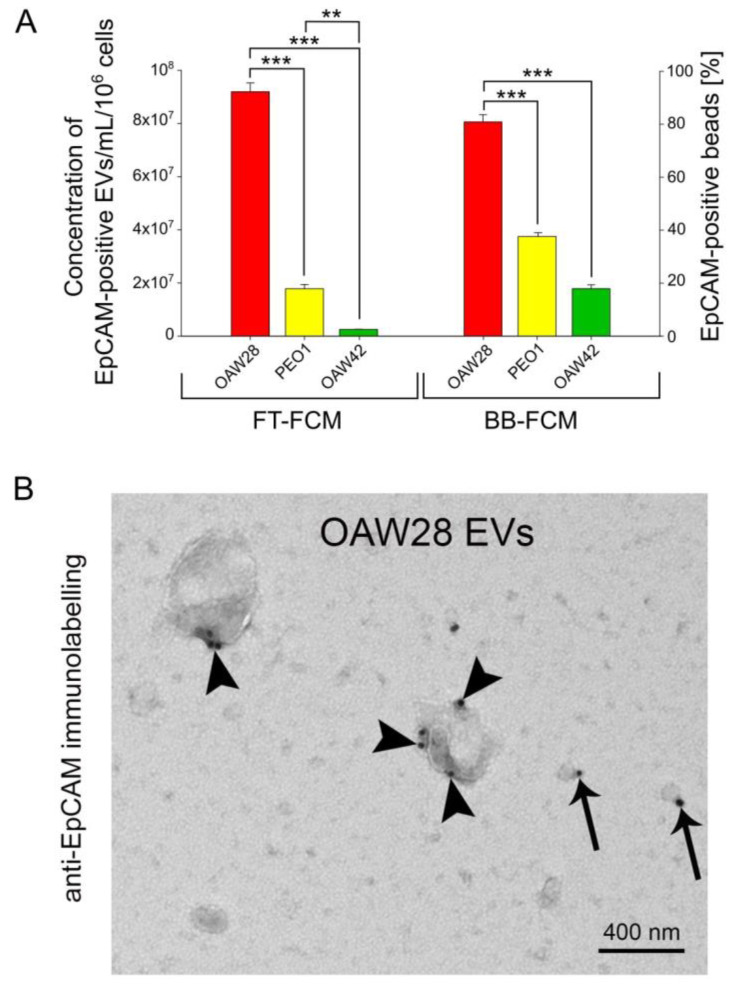
Concentration of EpCAM-positive extracellular vesicles (EVs) released in 24 h from OAW28 (red bar), PEO1 (yellow bar) and OAW42 (green bar) cells analyzed by fluorescence-triggered flow cytometry (left axis). Percentage of EpCAM-positive beads detected in isolated samples of EVs released in 24 h from OAW28 (red bar), PEO1 (yellow bar) and OAW42 (green bar) cells analyzed by bead-based flow cytometry (right axis) (**A**). Data are presented as mean ± SEM of three independent measurements made in duplicate. *** p* < 0.01; **** p* < 0.001. TEM micrograph showing the presence of EpCAM on the surface of large EVs (arrowheads) and small EVs (arrows) (**B**).

**Figure 6 pharmaceuticals-16-00907-f006:**
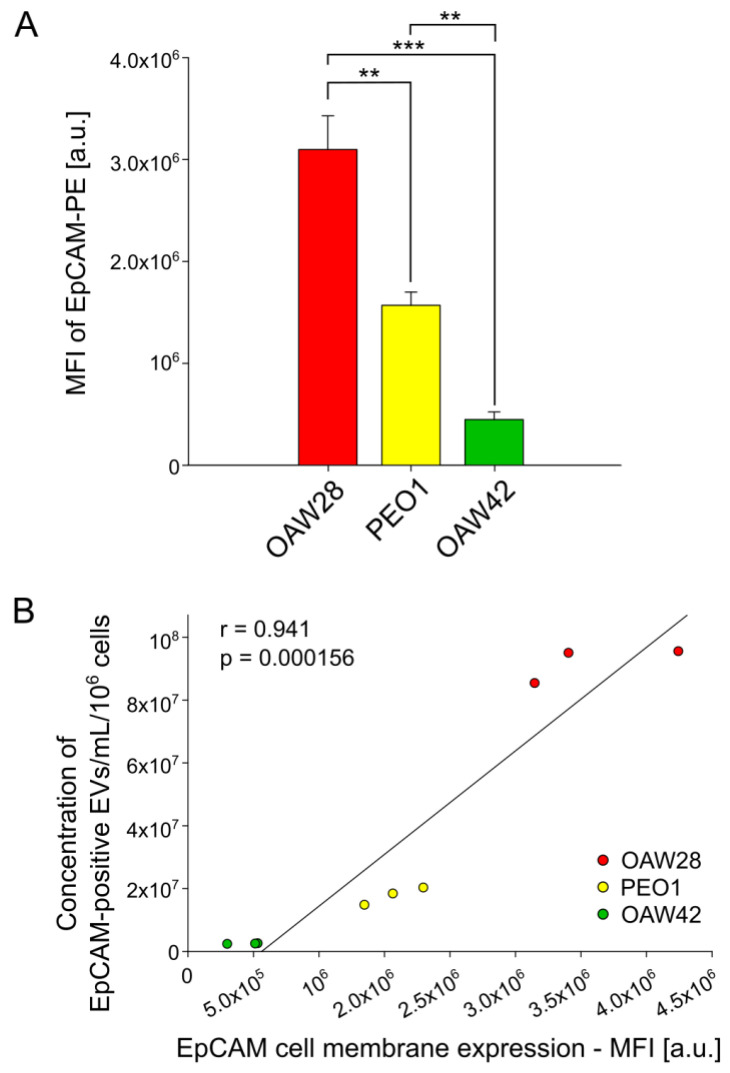
Expression of the extracellular EpCAM domain on OAW28 (red bar), PEO1 (yellow bar) and OAW42 (green bar) cells quantified by flow cytometry. MFI—mean fluorescent intensity. PE—phycoerythrin. Data are presented as mean ± SEM of three independent measurements made in duplicate. *** p* < 0.01; **** p* < 0.001. (**A**). Positive correlation between EpCAM expression on cell membrane and concentration of EpCAM-positive EVs released from OAW28, PEO1 and OAW42 cells (**B**). Data are presented as three independent measurements made in duplicate for each cell line.

## Data Availability

The data are contained within the article.

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
