# Peer review of "Characteristics of Extracellular Vesicles from a High-Grade Serous Ovarian Cancer Cell Line Derived from a Platinum-Resistant Patient as a Potential Tool for Aiding the Prediction of Responses to Chemotherapy"

_pharmaceuticals, 2023, doi:10.3390/ph16060907_

Round 1

Reviewer 1 Report

Research Manuscript Title: Characteristics of extracellular vesicles from an HGSOC cell line derived from a platinum-resistant patient as a potential tool for aiding prediction of response to chemotherapy

Manuscript ID: Pharmaceuticals-2404869

Platinum-resistant Ovarian cancer(OC) is usually a fatal affliction. Hence, a vital issue in OC research is the development of novel strategies to overcome platinum resistance. Treatment for OC is therefore moving towards personalized therapy. But, selection criteria are not entirely clear, since at present there are no validated molecular predictive biomarkers for platinum resistance. The authors have determined a strong positive correlation between the concentration of EpCAM-positive EVs and expression of cellular EpCAM in a cancer cell line. These results may contribute to prediction of platinum resistance in the future, although they should first be substantiated in clinical scenarios.

Major comments

There are some issues which need to be addressed by the authors:

Comment #1:

The authors must discuss and analyze their findings in relation to the study by Chebouti et al.[1].

Comment #2:

English language needs some touching up. Many sentences are confusing, and do not lead to scientific meaning.

Chebouti I, Kuhlmann JD, Buderath P, Weber S, Wimberger P, Bokeloh Y, Hauch S, Kimmig R, Kasimir-Bauer S. ERCC1-expressing circulating tumor cells as a potential diagnostic tool for monitoring response to platinum-based chemotherapy and for predicting post-therapeutic outcome of ovarian cancer. Oncotarget. 2017 Apr 4;8(15):24303.

English language needs some touching up. Many sentences are confusing, and do not lead to scientific meaning.

Author Response

We have read and carefully considered all the comments. We would like to thank you for the constructive suggestions and we hope that our responses meet your expectations.

In the corrected version, we have highlighted (with ’tracked changes’) new sentences, rewritten parts and some other changes. Some of these changes were made because of the comments of another reviewer. 

Below we provide our point-by-point response to the concerns.

Point 1: The authors must discuss and analyze their findings in relation to the study by Chebouti et al.

Response 1: We thank the reviewer for this suggestion, which will certainly improve interest in the article for readers of this journal. The study by Chebouti et al. investigates circulating tumor cells, so I have included an explanation of the connection between CTC and EpCAM molecules in the introduction section (line 92-97). In the discussion section, I have included an extensive discussion on the findings of this study (line 413-434). CTC are mainly enriched based on EpCAM alone or on EpCAM with other cell surface proteins. There is a commercial reagent kit already available for enrichment of tumor cells from whole blood in OC and the detection of cancer-associated gene expression in enriched tumor cells. However, this product is currently not intended for the diagnosis, prevention or treatment of OC. Chebouti et al. used this commercial reagent (AdnaTest) and came to interesting results; namely, although CTCs were captured by a procedure that targets EpCAM and MUC-1 surface epitopes, EpCAM and MUC-1 mRNA seemed to be downregulated. One of the pros of EVs in comparison to CTC is that RNA, DNA and proteins are protected by the EV membrane from nucleases, proteases, fluctuation of pH and other environmental factors. Direct comparison of EpCAM protein and EpCAM RNA levels between CTCs and EVs preferentially from the same sample might therefore explain this discrepancy.

There has recently been an intensification of research examining the value of complementary liquid biopsies. At the end of the discussion, therefore, I have emphasized that a combination of EVs, CTC and cell free DNA analysis from the same blood sample may help in selecting more effective therapies (line 461).

Since the discussion on results of the study by Chebouti et. al needs a broader description of this type of analysis, I have added 7 new references:

  • Chebouti, I.; Kuhlmann J. D.; Buderath P.; Weber S.; Wimberger P.; Bokeloh Y.; Hauch S.; Kimmig R.; Kasimir-Bauer S. ERCC1-expressing circulating tumor cells as a potential diagnostic tool for monitoring response to platinum-based chemotherapy and for predicting post-therapeutic outcome of ovarian cancer. Oncotarget 2017, 8(15), 24303-24313.
  • Zhang, X.; Li, H.; Yu, X.; Li S.; Lei, Z.; Li, C.; Zhang, Q.; Han, Q.; Li, Y.; Zhang, K.; et al. Analysis of Circulating Tumor Cells in Ovarian Cancer and Their Clinical Value as a Biomarker. Cell Physiol Biochem. 2018, 48(5), 1983-1994.
  • Franken, A.; Kraemer, A.; Sicking, A.; Watolla, M.; Rivandi, M.; Yang, L.; Warfsmann, J:, Polzer, B. M.; Friedl, T. W. P.; Meier-Stiegen, F.; et al. Comparative analysis of EpCAM high-expressing and low-expressing circulating tumour cells with regard to their clonal relationship and clinical value. Br J Cancer. 2023, 128(9), 1742-1752.
  • Lowes, L. E.; Bratman, S. V.; Dittamore, R.; Done, S.; Kelley, S. O.; Mai, S.; Morin, R. D.; Wyatt, A. W.; Allan, A. L. Circulating Tumor Cells (CTC) and Cell-Free DNA (cfDNA) Workshop 2016: Scientific Opportunities and Logistics for Cancer Clinical Trial Incorporation. Int J Mol Sci. 2016, 17(9), 1505.
  • Kim, K. M.; Abdelmohsen, K., Mustapic, M.; Kapogiannis, D.; Gorospe, M. RNA in extracellular vesicles. Wiley Interdiscip Rev RNA 2017, 8(4), 10.
  • Keup, C.; Kimmig, R.; Kasimir-Bauer, S. Combinatorial Power of cfDNA, CTCs and EVs in Oncology. Diagnostics (Basel) 2022, 12(4), 870.
  • Nanou, A.; Miao, J.; Coumans, F. A. W.; Dolce, E. M.; Darga, E.; Barlow, W.; Smerage, J. B.; Paoletti, C.; Godwin, A. K.; Pusztai, L., et al. Tumor-Derived Extracellular Vesicles as Complementary Prognostic Factors to Circulating Tumor Cells in Metastatic Breast Cancer. JCO Precis Oncol. 2023, 7: e2200372.

Point 2: English language needs some touching up. Many sentences are confusing, and do not lead to scientific meaning.

Response 2: The manuscript has been read again by an experienced English language editor and corrections made as required. We hope that this has removed any confusions.

We have read and carefully considered all the comments. We would like to thank you for the constructive suggestions and we hope that our responses meet your expectations.

In the corrected version, we have highlighted (with ’tracked changes’) new sentences, rewritten parts and some other changes. Some of these changes were made because of the comments of another reviewer. 

Below we provide our point-by-point response to the concerns.

Point 1: The authors must discuss and analyze their findings in relation to the study by Chebouti et al.

Response 1: We thank the reviewer for this suggestion, which will certainly improve interest in the article for readers of this journal. The study by Chebouti et al. investigates circulating tumor cells, so I have included an explanation of the connection between CTC and EpCAM molecules in the introduction section (line 92-97). In the discussion section, I have included an extensive discussion on the findings of this study (line 413-434). CTC are mainly enriched based on EpCAM alone or on EpCAM with other cell surface proteins. There is a commercial reagent kit already available for enrichment of tumor cells from whole blood in OC and the detection of cancer-associated gene expression in enriched tumor cells. However, this product is currently not intended for the diagnosis, prevention or treatment of OC. Chebouti et al. used this commercial reagent (AdnaTest) and came to interesting results; namely, although CTCs were captured by a procedure that targets EpCAM and MUC-1 surface epitopes, EpCAM and MUC-1 mRNA seemed to be downregulated. One of the pros of EVs in comparison to CTC is that RNA, DNA and proteins are protected by the EV membrane from nucleases, proteases, fluctuation of pH and other environmental factors. Direct comparison of EpCAM protein and EpCAM RNA levels between CTCs and EVs preferentially from the same sample might therefore explain this discrepancy.

There has recently been an intensification of research examining the value of complementary liquid biopsies. At the end of the discussion, therefore, I have emphasized that a combination of EVs, CTC and cell free DNA analysis from the same blood sample may help in selecting more effective therapies (line 461).

Since the discussion on results of the study by Chebouti et. al needs a broader description of this type of analysis, I have added 7 new references:

  • Chebouti, I.; Kuhlmann J. D.; Buderath P.; Weber S.; Wimberger P.; Bokeloh Y.; Hauch S.; Kimmig R.; Kasimir-Bauer S. ERCC1-expressing circulating tumor cells as a potential diagnostic tool for monitoring response to platinum-based chemotherapy and for predicting post-therapeutic outcome of ovarian cancer. Oncotarget 2017, 8(15), 24303-24313.
  • Zhang, X.; Li, H.; Yu, X.; Li S.; Lei, Z.; Li, C.; Zhang, Q.; Han, Q.; Li, Y.; Zhang, K.; et al. Analysis of Circulating Tumor Cells in Ovarian Cancer and Their Clinical Value as a Biomarker. Cell Physiol Biochem. 2018, 48(5), 1983-1994.
  • Franken, A.; Kraemer, A.; Sicking, A.; Watolla, M.; Rivandi, M.; Yang, L.; Warfsmann, J:, Polzer, B. M.; Friedl, T. W. P.; Meier-Stiegen, F.; et al. Comparative analysis of EpCAM high-expressing and low-expressing circulating tumour cells with regard to their clonal relationship and clinical value. Br J Cancer. 2023, 128(9), 1742-1752.
  • Lowes, L. E.; Bratman, S. V.; Dittamore, R.; Done, S.; Kelley, S. O.; Mai, S.; Morin, R. D.; Wyatt, A. W.; Allan, A. L. Circulating Tumor Cells (CTC) and Cell-Free DNA (cfDNA) Workshop 2016: Scientific Opportunities and Logistics for Cancer Clinical Trial Incorporation. Int J Mol Sci. 2016, 17(9), 1505.
  • Kim, K. M.; Abdelmohsen, K., Mustapic, M.; Kapogiannis, D.; Gorospe, M. RNA in extracellular vesicles. Wiley Interdiscip Rev RNA 2017, 8(4), 10.
  • Keup, C.; Kimmig, R.; Kasimir-Bauer, S. Combinatorial Power of cfDNA, CTCs and EVs in Oncology. Diagnostics (Basel) 2022, 12(4), 870.
  • Nanou, A.; Miao, J.; Coumans, F. A. W.; Dolce, E. M.; Darga, E.; Barlow, W.; Smerage, J. B.; Paoletti, C.; Godwin, A. K.; Pusztai, L., et al. Tumor-Derived Extracellular Vesicles as Complementary Prognostic Factors to Circulating Tumor Cells in Metastatic Breast Cancer. JCO Precis Oncol. 2023, 7: e2200372.

Point 2: English language needs some touching up. Many sentences are confusing, and do not lead to scientific meaning.

Response 2: The manuscript has been read again by an experienced English language editor and corrections made as required. We hope that this has removed any confusions.

We have read and carefully considered all the comments. We would like to thank you for the constructive suggestions and we hope that our responses meet your expectations.

In the corrected version, we have highlighted (with ’tracked changes’) new sentences, rewritten parts and some other changes. Some of these changes were made because of the comments of another reviewer. 

Below we provide our point-by-point response to the concerns.

Point 1: The authors must discuss and analyze their findings in relation to the study by Chebouti et al.

Response 1: We thank the reviewer for this suggestion, which will certainly improve interest in the article for readers of this journal. The study by Chebouti et al. investigates circulating tumor cells, so I have included an explanation of the connection between CTC and EpCAM molecules in the introduction section (line 92-97). In the discussion section, I have included an extensive discussion on the findings of this study (line 413-434). CTC are mainly enriched based on EpCAM alone or on EpCAM with other cell surface proteins. There is a commercial reagent kit already available for enrichment of tumor cells from whole blood in OC and the detection of cancer-associated gene expression in enriched tumor cells. However, this product is currently not intended for the diagnosis, prevention or treatment of OC. Chebouti et al. used this commercial reagent (AdnaTest) and came to interesting results; namely, although CTCs were captured by a procedure that targets EpCAM and MUC-1 surface epitopes, EpCAM and MUC-1 mRNA seemed to be downregulated. One of the pros of EVs in comparison to CTC is that RNA, DNA and proteins are protected by the EV membrane from nucleases, proteases, fluctuation of pH and other environmental factors. Direct comparison of EpCAM protein and EpCAM RNA levels between CTCs and EVs preferentially from the same sample might therefore explain this discrepancy.

There has recently been an intensification of research examining the value of complementary liquid biopsies. At the end of the discussion, therefore, I have emphasized that a combination of EVs, CTC and cell free DNA analysis from the same blood sample may help in selecting more effective therapies (line 461).

Since the discussion on results of the study by Chebouti et. al needs a broader description of this type of analysis, I have added 7 new references:

  • Chebouti, I.; Kuhlmann J. D.; Buderath P.; Weber S.; Wimberger P.; Bokeloh Y.; Hauch S.; Kimmig R.; Kasimir-Bauer S. ERCC1-expressing circulating tumor cells as a potential diagnostic tool for monitoring response to platinum-based chemotherapy and for predicting post-therapeutic outcome of ovarian cancer. Oncotarget 2017, 8(15), 24303-24313.
  • Zhang, X.; Li, H.; Yu, X.; Li S.; Lei, Z.; Li, C.; Zhang, Q.; Han, Q.; Li, Y.; Zhang, K.; et al. Analysis of Circulating Tumor Cells in Ovarian Cancer and Their Clinical Value as a Biomarker. Cell Physiol Biochem. 2018, 48(5), 1983-1994.
  • Franken, A.; Kraemer, A.; Sicking, A.; Watolla, M.; Rivandi, M.; Yang, L.; Warfsmann, J:, Polzer, B. M.; Friedl, T. W. P.; Meier-Stiegen, F.; et al. Comparative analysis of EpCAM high-expressing and low-expressing circulating tumour cells with regard to their clonal relationship and clinical value. Br J Cancer. 2023, 128(9), 1742-1752.
  • Lowes, L. E.; Bratman, S. V.; Dittamore, R.; Done, S.; Kelley, S. O.; Mai, S.; Morin, R. D.; Wyatt, A. W.; Allan, A. L. Circulating Tumor Cells (CTC) and Cell-Free DNA (cfDNA) Workshop 2016: Scientific Opportunities and Logistics for Cancer Clinical Trial Incorporation. Int J Mol Sci. 2016, 17(9), 1505.
  • Kim, K. M.; Abdelmohsen, K., Mustapic, M.; Kapogiannis, D.; Gorospe, M. RNA in extracellular vesicles. Wiley Interdiscip Rev RNA 2017, 8(4), 10.
  • Keup, C.; Kimmig, R.; Kasimir-Bauer, S. Combinatorial Power of cfDNA, CTCs and EVs in Oncology. Diagnostics (Basel) 2022, 12(4), 870.
  • Nanou, A.; Miao, J.; Coumans, F. A. W.; Dolce, E. M.; Darga, E.; Barlow, W.; Smerage, J. B.; Paoletti, C.; Godwin, A. K.; Pusztai, L., et al. Tumor-Derived Extracellular Vesicles as Complementary Prognostic Factors to Circulating Tumor Cells in Metastatic Breast Cancer. JCO Precis Oncol. 2023, 7: e2200372.

Point 2: English language needs some touching up. Many sentences are confusing, and do not lead to scientific meaning.

Response 2: The manuscript has been read again by an experienced English language editor and corrections made as required. We hope that this has removed any confusions.

We have read and carefully considered all the comments. We would like to thank you for the constructive suggestions and we hope that our responses meet your expectations.

In the corrected version, we have highlighted (with ’tracked changes’) new sentences, rewritten parts and some other changes. Some of these changes were made because of the comments of another reviewer. 

Below we provide our point-by-point response to the concerns.

Point 1: The authors must discuss and analyze their findings in relation to the study by Chebouti et al.

Response 1: We thank the reviewer for this suggestion, which will certainly improve interest in the article for readers of this journal. The study by Chebouti et al. investigates circulating tumor cells, so I have included an explanation of the connection between CTC and EpCAM molecules in the introduction section (line 92-97). In the discussion section, I have included an extensive discussion on the findings of this study (line 413-434). CTC are mainly enriched based on EpCAM alone or on EpCAM with other cell surface proteins. There is a commercial reagent kit already available for enrichment of tumor cells from whole blood in OC and the detection of cancer-associated gene expression in enriched tumor cells. However, this product is currently not intended for the diagnosis, prevention or treatment of OC. Chebouti et al. used this commercial reagent (AdnaTest) and came to interesting results; namely, although CTCs were captured by a procedure that targets EpCAM and MUC-1 surface epitopes, EpCAM and MUC-1 mRNA seemed to be downregulated. One of the pros of EVs in comparison to CTC is that RNA, DNA and proteins are protected by the EV membrane from nucleases, proteases, fluctuation of pH and other environmental factors. Direct comparison of EpCAM protein and EpCAM RNA levels between CTCs and EVs preferentially from the same sample might therefore explain this discrepancy.

There has recently been an intensification of research examining the value of complementary liquid biopsies. At the end of the discussion, therefore, I have emphasized that a combination of EVs, CTC and cell free DNA analysis from the same blood sample may help in selecting more effective therapies (line 461).

Since the discussion on results of the study by Chebouti et. al needs a broader description of this type of analysis, I have added 7 new references:

  • Chebouti, I.; Kuhlmann J. D.; Buderath P.; Weber S.; Wimberger P.; Bokeloh Y.; Hauch S.; Kimmig R.; Kasimir-Bauer S. ERCC1-expressing circulating tumor cells as a potential diagnostic tool for monitoring response to platinum-based chemotherapy and for predicting post-therapeutic outcome of ovarian cancer. Oncotarget 2017, 8(15), 24303-24313.
  • Zhang, X.; Li, H.; Yu, X.; Li S.; Lei, Z.; Li, C.; Zhang, Q.; Han, Q.; Li, Y.; Zhang, K.; et al. Analysis of Circulating Tumor Cells in Ovarian Cancer and Their Clinical Value as a Biomarker. Cell Physiol Biochem. 2018, 48(5), 1983-1994.
  • Franken, A.; Kraemer, A.; Sicking, A.; Watolla, M.; Rivandi, M.; Yang, L.; Warfsmann, J:, Polzer, B. M.; Friedl, T. W. P.; Meier-Stiegen, F.; et al. Comparative analysis of EpCAM high-expressing and low-expressing circulating tumour cells with regard to their clonal relationship and clinical value. Br J Cancer. 2023, 128(9), 1742-1752.
  • Lowes, L. E.; Bratman, S. V.; Dittamore, R.; Done, S.; Kelley, S. O.; Mai, S.; Morin, R. D.; Wyatt, A. W.; Allan, A. L. Circulating Tumor Cells (CTC) and Cell-Free DNA (cfDNA) Workshop 2016: Scientific Opportunities and Logistics for Cancer Clinical Trial Incorporation. Int J Mol Sci. 2016, 17(9), 1505.
  • Kim, K. M.; Abdelmohsen, K., Mustapic, M.; Kapogiannis, D.; Gorospe, M. RNA in extracellular vesicles. Wiley Interdiscip Rev RNA 2017, 8(4), 10.
  • Keup, C.; Kimmig, R.; Kasimir-Bauer, S. Combinatorial Power of cfDNA, CTCs and EVs in Oncology. Diagnostics (Basel) 2022, 12(4), 870.
  • Nanou, A.; Miao, J.; Coumans, F. A. W.; Dolce, E. M.; Darga, E.; Barlow, W.; Smerage, J. B.; Paoletti, C.; Godwin, A. K.; Pusztai, L., et al. Tumor-Derived Extracellular Vesicles as Complementary Prognostic Factors to Circulating Tumor Cells in Metastatic Breast Cancer. JCO Precis Oncol. 2023, 7: e2200372.

Point 2: English language needs some touching up. Many sentences are confusing, and do not lead to scientific meaning.

Response 2: The manuscript has been read again by an experienced English language editor and corrections made as required. We hope that this has removed any confusions.

We have read and carefully considered all the comments. We would like to thank you for the constructive suggestions and we hope that our responses meet your expectations.

In the corrected version, we have highlighted (with ’tracked changes’) new sentences, rewritten parts and some other changes. Some of these changes were made because of the comments of another reviewer. 

Below we provide our point-by-point response to the concerns.

Point 1: The authors must discuss and analyze their findings in relation to the study by Chebouti et al.

Response 1: We thank the reviewer for this suggestion, which will certainly improve interest in the article for readers of this journal. The study by Chebouti et al. investigates circulating tumor cells, so I have included an explanation of the connection between CTC and EpCAM molecules in the introduction section (line 92-97). In the discussion section, I have included an extensive discussion on the findings of this study (line 413-434). CTC are mainly enriched based on EpCAM alone or on EpCAM with other cell surface proteins. There is a commercial reagent kit already available for enrichment of tumor cells from whole blood in OC and the detection of cancer-associated gene expression in enriched tumor cells. However, this product is currently not intended for the diagnosis, prevention or treatment of OC. Chebouti et al. used this commercial reagent (AdnaTest) and came to interesting results; namely, although CTCs were captured by a procedure that targets EpCAM and MUC-1 surface epitopes, EpCAM and MUC-1 mRNA seemed to be downregulated. One of the pros of EVs in comparison to CTC is that RNA, DNA and proteins are protected by the EV membrane from nucleases, proteases, fluctuation of pH and other environmental factors. Direct comparison of EpCAM protein and EpCAM RNA levels between CTCs and EVs preferentially from the same sample might therefore explain this discrepancy.

There has recently been an intensification of research examining the value of complementary liquid biopsies. At the end of the discussion, therefore, I have emphasized that a combination of EVs, CTC and cell free DNA analysis from the same blood sample may help in selecting more effective therapies (line 461).

Since the discussion on results of the study by Chebouti et. al needs a broader description of this type of analysis, I have added 7 new references:

  • Chebouti, I.; Kuhlmann J. D.; Buderath P.; Weber S.; Wimberger P.; Bokeloh Y.; Hauch S.; Kimmig R.; Kasimir-Bauer S. ERCC1-expressing circulating tumor cells as a potential diagnostic tool for monitoring response to platinum-based chemotherapy and for predicting post-therapeutic outcome of ovarian cancer. Oncotarget 2017, 8(15), 24303-24313.
  • Zhang, X.; Li, H.; Yu, X.; Li S.; Lei, Z.; Li, C.; Zhang, Q.; Han, Q.; Li, Y.; Zhang, K.; et al. Analysis of Circulating Tumor Cells in Ovarian Cancer and Their Clinical Value as a Biomarker. Cell Physiol Biochem. 2018, 48(5), 1983-1994.
  • Franken, A.; Kraemer, A.; Sicking, A.; Watolla, M.; Rivandi, M.; Yang, L.; Warfsmann, J:, Polzer, B. M.; Friedl, T. W. P.; Meier-Stiegen, F.; et al. Comparative analysis of EpCAM high-expressing and low-expressing circulating tumour cells with regard to their clonal relationship and clinical value. Br J Cancer. 2023, 128(9), 1742-1752.
  • Lowes, L. E.; Bratman, S. V.; Dittamore, R.; Done, S.; Kelley, S. O.; Mai, S.; Morin, R. D.; Wyatt, A. W.; Allan, A. L. Circulating Tumor Cells (CTC) and Cell-Free DNA (cfDNA) Workshop 2016: Scientific Opportunities and Logistics for Cancer Clinical Trial Incorporation. Int J Mol Sci. 2016, 17(9), 1505.
  • Kim, K. M.; Abdelmohsen, K., Mustapic, M.; Kapogiannis, D.; Gorospe, M. RNA in extracellular vesicles. Wiley Interdiscip Rev RNA 2017, 8(4), 10.
  • Keup, C.; Kimmig, R.; Kasimir-Bauer, S. Combinatorial Power of cfDNA, CTCs and EVs in Oncology. Diagnostics (Basel) 2022, 12(4), 870.
  • Nanou, A.; Miao, J.; Coumans, F. A. W.; Dolce, E. M.; Darga, E.; Barlow, W.; Smerage, J. B.; Paoletti, C.; Godwin, A. K.; Pusztai, L., et al. Tumor-Derived Extracellular Vesicles as Complementary Prognostic Factors to Circulating Tumor Cells in Metastatic Breast Cancer. JCO Precis Oncol. 2023, 7: e2200372.

Point 2: English language needs some touching up. Many sentences are confusing, and do not lead to scientific meaning.

Response 2: The manuscript has been read again by an experienced English language editor and corrections made as required. We hope that this has removed any confusions.

We have read and carefully considered all the comments. We would like to thank you for the constructive suggestions and we hope that our responses meet your expectations.

In the corrected version, we have highlighted (with ’tracked changes’) new sentences, rewritten parts and some other changes. Some of these changes were made because of the comments of another reviewer. 

Below we provide our point-by-point response to the concerns.

Point 1: The authors must discuss and analyze their findings in relation to the study by Chebouti et al.

Response 1: We thank the reviewer for this suggestion, which will certainly improve interest in the article for readers of this journal. The study by Chebouti et al. investigates circulating tumor cells, so I have included an explanation of the connection between CTC and EpCAM molecules in the introduction section (line 92-97). In the discussion section, I have included an extensive discussion on the findings of this study (line 413-434). CTC are mainly enriched based on EpCAM alone or on EpCAM with other cell surface proteins. There is a commercial reagent kit already available for enrichment of tumor cells from whole blood in OC and the detection of cancer-associated gene expression in enriched tumor cells. However, this product is currently not intended for the diagnosis, prevention or treatment of OC. Chebouti et al. used this commercial reagent (AdnaTest) and came to interesting results; namely, although CTCs were captured by a procedure that targets EpCAM and MUC-1 surface epitopes, EpCAM and MUC-1 mRNA seemed to be downregulated. One of the pros of EVs in comparison to CTC is that RNA, DNA and proteins are protected by the EV membrane from nucleases, proteases, fluctuation of pH and other environmental factors. Direct comparison of EpCAM protein and EpCAM RNA levels between CTCs and EVs preferentially from the same sample might therefore explain this discrepancy.

There has recently been an intensification of research examining the value of complementary liquid biopsies. At the end of the discussion, therefore, I have emphasized that a combination of EVs, CTC and cell free DNA analysis from the same blood sample may help in selecting more effective therapies (line 461).

Since the discussion on results of the study by Chebouti et. al needs a broader description of this type of analysis, I have added 7 new references:

  • Chebouti, I.; Kuhlmann J. D.; Buderath P.; Weber S.; Wimberger P.; Bokeloh Y.; Hauch S.; Kimmig R.; Kasimir-Bauer S. ERCC1-expressing circulating tumor cells as a potential diagnostic tool for monitoring response to platinum-based chemotherapy and for predicting post-therapeutic outcome of ovarian cancer. Oncotarget 2017, 8(15), 24303-24313.
  • Zhang, X.; Li, H.; Yu, X.; Li S.; Lei, Z.; Li, C.; Zhang, Q.; Han, Q.; Li, Y.; Zhang, K.; et al. Analysis of Circulating Tumor Cells in Ovarian Cancer and Their Clinical Value as a Biomarker. Cell Physiol Biochem. 2018, 48(5), 1983-1994.
  • Franken, A.; Kraemer, A.; Sicking, A.; Watolla, M.; Rivandi, M.; Yang, L.; Warfsmann, J:, Polzer, B. M.; Friedl, T. W. P.; Meier-Stiegen, F.; et al. Comparative analysis of EpCAM high-expressing and low-expressing circulating tumour cells with regard to their clonal relationship and clinical value. Br J Cancer. 2023, 128(9), 1742-1752.
  • Lowes, L. E.; Bratman, S. V.; Dittamore, R.; Done, S.; Kelley, S. O.; Mai, S.; Morin, R. D.; Wyatt, A. W.; Allan, A. L. Circulating Tumor Cells (CTC) and Cell-Free DNA (cfDNA) Workshop 2016: Scientific Opportunities and Logistics for Cancer Clinical Trial Incorporation. Int J Mol Sci. 2016, 17(9), 1505.
  • Kim, K. M.; Abdelmohsen, K., Mustapic, M.; Kapogiannis, D.; Gorospe, M. RNA in extracellular vesicles. Wiley Interdiscip Rev RNA 2017, 8(4), 10.
  • Keup, C.; Kimmig, R.; Kasimir-Bauer, S. Combinatorial Power of cfDNA, CTCs and EVs in Oncology. Diagnostics (Basel) 2022, 12(4), 870.
  • Nanou, A.; Miao, J.; Coumans, F. A. W.; Dolce, E. M.; Darga, E.; Barlow, W.; Smerage, J. B.; Paoletti, C.; Godwin, A. K.; Pusztai, L., et al. Tumor-Derived Extracellular Vesicles as Complementary Prognostic Factors to Circulating Tumor Cells in Metastatic Breast Cancer. JCO Precis Oncol. 2023, 7: e2200372.

Point 2: English language needs some touching up. Many sentences are confusing, and do not lead to scientific meaning.

Response 2: The manuscript has been read again by an experienced English language editor and corrections made as required. We hope that this has removed any confusions.

We have read and carefully considered all the comments. We would like to thank you for the constructive suggestions and we hope that our responses meet your expectations.

In the corrected version, we have highlighted (with ’tracked changes’) new sentences, rewritten parts and some other changes. Some of these changes were made because of the comments of another reviewer. 

Below we provide our point-by-point response to the concerns.

Point 1: The authors must discuss and analyze their findings in relation to the study by Chebouti et al.

Response 1: We thank the reviewer for this suggestion, which will certainly improve interest in the article for readers of this journal. The study by Chebouti et al. investigates circulating tumor cells, so I have included an explanation of the connection between CTC and EpCAM molecules in the introduction section (line 92-97). In the discussion section, I have included an extensive discussion on the findings of this study (line 413-434). CTC are mainly enriched based on EpCAM alone or on EpCAM with other cell surface proteins. There is a commercial reagent kit already available for enrichment of tumor cells from whole blood in OC and the detection of cancer-associated gene expression in enriched tumor cells. However, this product is currently not intended for the diagnosis, prevention or treatment of OC. Chebouti et al. used this commercial reagent (AdnaTest) and came to interesting results; namely, although CTCs were captured by a procedure that targets EpCAM and MUC-1 surface epitopes, EpCAM and MUC-1 mRNA seemed to be downregulated. One of the pros of EVs in comparison to CTC is that RNA, DNA and proteins are protected by the EV membrane from nucleases, proteases, fluctuation of pH and other environmental factors. Direct comparison of EpCAM protein and EpCAM RNA levels between CTCs and EVs preferentially from the same sample might therefore explain this discrepancy.

There has recently been an intensification of research examining the value of complementary liquid biopsies. At the end of the discussion, therefore, I have emphasized that a combination of EVs, CTC and cell free DNA analysis from the same blood sample may help in selecting more effective therapies (line 461).

Since the discussion on results of the study by Chebouti et. al needs a broader description of this type of analysis, I have added 7 new references:

  • Chebouti, I.; Kuhlmann J. D.; Buderath P.; Weber S.; Wimberger P.; Bokeloh Y.; Hauch S.; Kimmig R.; Kasimir-Bauer S. ERCC1-expressing circulating tumor cells as a potential diagnostic tool for monitoring response to platinum-based chemotherapy and for predicting post-therapeutic outcome of ovarian cancer. Oncotarget 2017, 8(15), 24303-24313.
  • Zhang, X.; Li, H.; Yu, X.; Li S.; Lei, Z.; Li, C.; Zhang, Q.; Han, Q.; Li, Y.; Zhang, K.; et al. Analysis of Circulating Tumor Cells in Ovarian Cancer and Their Clinical Value as a Biomarker. Cell Physiol Biochem. 2018, 48(5), 1983-1994.
  • Franken, A.; Kraemer, A.; Sicking, A.; Watolla, M.; Rivandi, M.; Yang, L.; Warfsmann, J:, Polzer, B. M.; Friedl, T. W. P.; Meier-Stiegen, F.; et al. Comparative analysis of EpCAM high-expressing and low-expressing circulating tumour cells with regard to their clonal relationship and clinical value. Br J Cancer. 2023, 128(9), 1742-1752.
  • Lowes, L. E.; Bratman, S. V.; Dittamore, R.; Done, S.; Kelley, S. O.; Mai, S.; Morin, R. D.; Wyatt, A. W.; Allan, A. L. Circulating Tumor Cells (CTC) and Cell-Free DNA (cfDNA) Workshop 2016: Scientific Opportunities and Logistics for Cancer Clinical Trial Incorporation. Int J Mol Sci. 2016, 17(9), 1505.
  • Kim, K. M.; Abdelmohsen, K., Mustapic, M.; Kapogiannis, D.; Gorospe, M. RNA in extracellular vesicles. Wiley Interdiscip Rev RNA 2017, 8(4), 10.
  • Keup, C.; Kimmig, R.; Kasimir-Bauer, S. Combinatorial Power of cfDNA, CTCs and EVs in Oncology. Diagnostics (Basel) 2022, 12(4), 870.
  • Nanou, A.; Miao, J.; Coumans, F. A. W.; Dolce, E. M.; Darga, E.; Barlow, W.; Smerage, J. B.; Paoletti, C.; Godwin, A. K.; Pusztai, L., et al. Tumor-Derived Extracellular Vesicles as Complementary Prognostic Factors to Circulating Tumor Cells in Metastatic Breast Cancer. JCO Precis Oncol. 2023, 7: e2200372.

Point 2: English language needs some touching up. Many sentences are confusing, and do not lead to scientific meaning.

Response 2: The manuscript has been read again by an experienced English language editor and corrections made as required. We hope that this has removed any confusions.

We have read and carefully considered all the comments. We would like to thank you for the constructive suggestions and we hope that our responses meet your expectations.

In the corrected version, we have highlighted (with ’tracked changes’) new sentences, rewritten parts and some other changes. Some of these changes were made because of the comments of another reviewer. 

Below we provide our point-by-point response to the concerns.

Point 1: The authors must discuss and analyze their findings in relation to the study by Chebouti et al.

Response 1: We thank the reviewer for this suggestion, which will certainly improve interest in the article for readers of this journal. The study by Chebouti et al. investigates circulating tumor cells, so I have included an explanation of the connection between CTC and EpCAM molecules in the introduction section (line 92-97). In the discussion section, I have included an extensive discussion on the findings of this study (line 413-434). CTC are mainly enriched based on EpCAM alone or on EpCAM with other cell surface proteins. There is a commercial reagent kit already available for enrichment of tumor cells from whole blood in OC and the detection of cancer-associated gene expression in enriched tumor cells. However, this product is currently not intended for the diagnosis, prevention or treatment of OC. Chebouti et al. used this commercial reagent (AdnaTest) and came to interesting results; namely, although CTCs were captured by a procedure that targets EpCAM and MUC-1 surface epitopes, EpCAM and MUC-1 mRNA seemed to be downregulated. One of the pros of EVs in comparison to CTC is that RNA, DNA and proteins are protected by the EV membrane from nucleases, proteases, fluctuation of pH and other environmental factors. Direct comparison of EpCAM protein and EpCAM RNA levels between CTCs and EVs preferentially from the same sample might therefore explain this discrepancy.

There has recently been an intensification of research examining the value of complementary liquid biopsies. At the end of the discussion, therefore, I have emphasized that a combination of EVs, CTC and cell free DNA analysis from the same blood sample may help in selecting more effective therapies (line 461).

Since the discussion on results of the study by Chebouti et. al needs a broader description of this type of analysis, I have added 7 new references:

  • Chebouti, I.; Kuhlmann J. D.; Buderath P.; Weber S.; Wimberger P.; Bokeloh Y.; Hauch S.; Kimmig R.; Kasimir-Bauer S. ERCC1-expressing circulating tumor cells as a potential diagnostic tool for monitoring response to platinum-based chemotherapy and for predicting post-therapeutic outcome of ovarian cancer. Oncotarget 2017, 8(15), 24303-24313.
  • Zhang, X.; Li, H.; Yu, X.; Li S.; Lei, Z.; Li, C.; Zhang, Q.; Han, Q.; Li, Y.; Zhang, K.; et al. Analysis of Circulating Tumor Cells in Ovarian Cancer and Their Clinical Value as a Biomarker. Cell Physiol Biochem. 2018, 48(5), 1983-1994.
  • Franken, A.; Kraemer, A.; Sicking, A.; Watolla, M.; Rivandi, M.; Yang, L.; Warfsmann, J:, Polzer, B. M.; Friedl, T. W. P.; Meier-Stiegen, F.; et al. Comparative analysis of EpCAM high-expressing and low-expressing circulating tumour cells with regard to their clonal relationship and clinical value. Br J Cancer. 2023, 128(9), 1742-1752.
  • Lowes, L. E.; Bratman, S. V.; Dittamore, R.; Done, S.; Kelley, S. O.; Mai, S.; Morin, R. D.; Wyatt, A. W.; Allan, A. L. Circulating Tumor Cells (CTC) and Cell-Free DNA (cfDNA) Workshop 2016: Scientific Opportunities and Logistics for Cancer Clinical Trial Incorporation. Int J Mol Sci. 2016, 17(9), 1505.
  • Kim, K. M.; Abdelmohsen, K., Mustapic, M.; Kapogiannis, D.; Gorospe, M. RNA in extracellular vesicles. Wiley Interdiscip Rev RNA 2017, 8(4), 10.
  • Keup, C.; Kimmig, R.; Kasimir-Bauer, S. Combinatorial Power of cfDNA, CTCs and EVs in Oncology. Diagnostics (Basel) 2022, 12(4), 870.
  • Nanou, A.; Miao, J.; Coumans, F. A. W.; Dolce, E. M.; Darga, E.; Barlow, W.; Smerage, J. B.; Paoletti, C.; Godwin, A. K.; Pusztai, L., et al. Tumor-Derived Extracellular Vesicles as Complementary Prognostic Factors to Circulating Tumor Cells in Metastatic Breast Cancer. JCO Precis Oncol. 2023, 7: e2200372.

Point 2: English language needs some touching up. Many sentences are confusing, and do not lead to scientific meaning.

Response 2: The manuscript has been read again by an experienced English language editor and corrections made as required. We hope that this has removed any confusions.

Reviewer 2 Report

Article of Cerne et al is very interesting that have explore the potential of EpCAM specific EVs in predicting the platinum resistance cancer. Though results are convincing, it will be good to have two cell lines to compare the results.

Author Response

We would like to thank the referee for their constructive suggestions

Point 1: Though results are convincing, it will be good to have two cell lines to compare the results.

Response 1: We completely agree with your concern. However, we found only one clinically confirmed platinum-resistant HGSOC cell line.

Reviewer 3 Report

Dear Editor, Dear Authors,

Thank you for providing me with the chance to review this paper.

The manuscript is well-thought-out and well-described. The abstract clearly states the issue, and the introduction appropriately introduces the reader to the project's subject. Cited references are relevant to the research.

Minor:

In the introduction, it would be worthwhile to include the percentage of EpCAM expression in HGSOC, which might be an essential limit in diagnostics.

Lines 39-42, It is worth noting that, despite molecule-targeted pharmaceuticals, there has been a marginal improvement in the overall survival rate of OC patients since the introduction of cisplatin.

Once utilized, the acronym (ovarian cancer, OC) should be used throughout the text.

The abbreviation EV should be explained in the captions of the figures - the figure can operate independently of the text.

Line 142 consists of a dot in the middle of a sentence.

mL is the suitable unit, not ml

p for statistics should be in italics

The subject of the content of EV with EpCAM expression is intriguing; the previously described exosome with miR21 content has been linked to paclitaxel resistance (Au Yeung et al. 2016).

Author Response

We have read and carefully considered all the comments. We would like to thank you for the constructive suggestions and we hope that our responses meet your expectations.

In the corrected version, we have highlighted (with ’tracked changes’) new sentences, rewritten parts and some other changes. Some of these changes were made because of the comments of another reviewer.  

Below we provide our point-by-point response to your comments and concerns.

Point 1: In the introduction, it would be worthwhile to include the percentage of EpCAM expression in HGSOC, which might be an essential limit in diagnostics.

Response 1: We agree with this proposal and have included the percentage of EpCAM overexpression in serous OC in the introduction (line 92-93). We could not state data for HGSOC, since studies (Spizzo 2006, Woopen 2014) were published before the dualistic model proposed by Kurman and Shih was officially recognized in 2014 by the WHO. The authors therefore did not state their results for HGSOC.

Point 2: Lines 39-42, It is worth noting that, despite molecule-targeted pharmaceuticals, there has been a marginal improvement in the overall survival rate of OC patients since the introduction of cisplatin.

Response 2: We agree with this proposal and have included a sentence in the first paragraph of the introduction (line 50-53) and added the corresponding reference, Garrido et al (reference 5).

Point 3: Once utilized, the acronym (ovarian cancer, OC) should be used throughout the text.

Response 3: We have replaced ovarian cancer (lines: 108, 246, 436) with the acronym OC.

Point 4: The abbreviation EV should be explained in the captions of the figures - the figure can operate independently of the text.

Response 4: We have included the full term in the first mention  in the captions of all figures.

Point 5: Line 142 consists of a dot in the middle of a sentence.

Response 5: We have deleted the dot in the middle of the sentence.

Point 6: mL is the suitable unit, not ml

Response 6: We have replaced ml with mL throughout the manuscript.

Point 7: p for statistics should be in italics.

Response 7: We have corrected this mistake.

Point 8: The subject of the content of EV with EpCAM expression is intriguing; the previously described exosome with miR21 content has been linked to paclitaxel resistance (Au Yeung et al. 2016).

Response 8: We agree that the subject of the content of EV with EpCAM expression is intriguing and have included a new sentence in the discussion, on the association of miR-21 with paclitaxel resistance (line 405-406) and have added the corresponding reference, Yeung et al (reference 54).

MiR21 is one of the best studied miRNAs and is upregulated in almost all human cancers. MiR-21 targets various genes involved in many pathways that can justify chemoresistance. However, interpretation of studies on miRNAs is complicated, since the pre-miRNA is exported to cytoplasm, where double-stranded miRNA is cleaved to a mature strand and passenger strand. Alharbi et al (2020) found that overexpression of passenger strand miR-21-3p in EVs of OC cells is associated with resistance to cisplatin. Since Taylor et al in their study performed an analysis with a reagent that is capable of detecting only mature miRNAs, the content of miR-21-3p was probably not checked.

Round 2

Reviewer 1 Report

All concerns raised have been addressed.  Article now fit for publication.

Minor editing of English language required.